

# A study of salivary cortisol and glutamate after the cold pressor task in healthy adults

Roxaneh Zarnegar[1,2], Angeliki Vounta[2], Arisara Amrapala[2,3] and Sara S. Ghoreishizadeh[2,4]

[1] Royal National Orthopaedic Hospital NHS Trust, Stanmore, United Kingdom
[2] Institute of Orthopaedics and Musculoskeletal Science, University College London, University of London, Stanmore, United Kingdom
[3] Department of Psychiatry, Faculty of Medicine, Chulalongkorn University, Bangkok, Thailand
[4] Department of Electronic and Electrical Engineering, University College London, University of London, London, United Kingdom

Corresponding author
Sara S. Ghoreishizadeh,
s.ghoreishizadeh@ucl.ac.uk

## ABSTRACT

Nociception related salivary biomolecules can be a useful future aid in the assessment of acute pain. We have investigated changes in the levels of two salivary biomolecules, glutamate and cortisol, following the induction of acute cold pain using the cold pressor task (CPT). Saliva samples were collected from 18 healthy volunteers before, immediately after and then, every 10 minutes for one hour after CPT. Statistical analysis of the biomolecule concentrations across all participants and time points were done. This showed significant differences between salivary cortisol concentration before (median 0.14 µg/dL, Interquartile Range (IQR) = 0.1) and 10 minutes after termination of CPT (median 0.34 µg/dL, IQR = 0.4, $p = 0.007$). Male participants exhibited a greater increase in cortisol concentration after cold pain compared to females. The timeline and pattern of the rise in salivary cortisol concentration in this study are consistent with existing literature. Salivary glutamate concentration fluctuated but none of the changes were statistically significant except at $t = +50$ minutes, when the concentration had dropped below baseline. The findings do not support the use of glutamate as a useful biomarker in acute pain despite evidence that plasma and salivary glutamate levels are higher in people with chronic pain conditions such as migraine and temporomandibular disorder.

## INTRODUCTION

Pain is a subjective experience and its measurement is inherently reliant on self-report (*Raja et al., 2020*). In the treatment of acute pain, this is problematic when patients are unable to communicate their pain, for example when they are anaesthetized. Nonetheless, optimal treatment of pain under these circumstances is of paramount importance. In such circumstances, biomarkers that provide a measure for nociception would be useful to help healthcare professionals titrate analgesia requirements.

Saliva has vital functions in digestion, taste sensation and oral health (*Dawes et al., 2015*). Further to this, numerous biomolecules that have a role in physiological functions enter the saliva from blood by diffusion or active transport. Technologies (*Dongiovanni et al., 2023*) that enable salivary biomolecule measurement are progressing rapidly. Compared with other biofluids such as blood or cerebrospinal fluid, saliva can be collected with minimal discomfort and no specific training is needed for the collection process. Finding a salivary biomarker or a panel of biomarkers that signal nociception and can be used to determine analgesia requirements would be useful in both clinical practice and research.

Salivary levels of cortisol, a crucial component of the neuroendocrine hypothalamic-pituitary-adrenal (HPA) axis response to physiological stress (such as pain), are an indicator of the active unbound cortisol level in plasma (*Dorn et al., 2007*). Free serum cortisol diffuses passively into the saliva with a reasonably good correlation between serum and salivary concentration, independent of salivary flow (*El-Farhan, Rees & Evans, 2017*; *Bastin, Maiter & Gruson, 2018*). Salivary cortisol is known to change after cold pressor task (CPT) with a well-recognised time course (*Zarnegar et al., 2024*).

Glutamate is the main excitatory neurotransmitter in the human nervous system and is an essential biomolecule in nociceptive pathways (*Temmermand, Barrett & Fontana, 2022*), released both centrally and in the peripheral tissues in response to noxious stimuli (*Miller et al., 2011*). Current evidence shows that people with migraine have higher plasma and salivary glutamate levels during headache-free periods compared with healthy controls (*Demartini et al., 2023*). A study comparing plasma and salivary levels of glutamate in people with temporomandibular disorder has yielded similar results (*Jasim et al., 2020*). Therefore, there is evidence that salivary glutamate rises in conditions where pain is chronic, that is, when it has been present for over 3 months or when it persists beyond the period of healing after injury. In contrast, this salivary biomolecule has not been studied in the context of acute pain, that is, pain immediately after actual or potential tissue injury. Acute pain detection and treatment is integral to the care of people after surgery and trauma. Finding salivary biomarkers that can help in this respect would be greatly helpful to clinicians. In this study, as a step towards addressing the knowledge gap about change in salivary glutamate with acute pain, we have explored this after controlled exposure to acute cold pain using the cold pressor task in healthy human volunteers. While there is no recognised biological relationship between salivary cortisol and glutamate, we chose to measure them together on the same saliva samples with the rationale that detecting the already known pattern of change in cortisol will serve to validate the method and conduct of our experiments, strengthening our conclusions on change in salivary glutamate.

## METHOD

The study was approved by the UCL Research Ethics Committee (Ref No. 15021/001) and complied with the International Ethical Guidelines for Biomedical Research Involving Human Subjects, Good Clinical Practice guidelines, and the Declaration of Helsinki. All subjects provided written informed consent prior to their participation.

## Participants
### Recruitment

Staff members within the UCL Institute of Orthopaedics and Musculoskeletal Science and the Royal National Orthopaedic Hospital, Stanmore, UK were invited to participate by internal staff email. An information sheet was sent out to explain the aim of the study and familiarise potential participants with the protocol. Interested individuals were asked to contact the research team. All participants received £10 as reasonable expenses for taking part in the study.

Inclusion criteria were age range 18–70 years old, overall good health, and ability to maintain good oral hygiene on the day of the experiment. Good health was defined by participant self-report, on direct questioning, that they feel well and require no regular medications. Exclusion criteria were pre-existing pain, pregnancy or lactation, poor dental health, tobacco use, regular use of medication (including oral contraceptives and antidepressants) or diagnosed systemic muscular or joint diseases, neurological disorders, anxiety or depression, or high blood pressure.

### Sample size

Rise in cortisol concentration after CPT has been demonstrable in studies with 10–20 participants (*Goodin et al., 2012*; *Finke et al., 2021*). In addition, using data in previous research on salivary cortisol change after pain induction (*Finke et al., 2021*), sample size calculation shows that in order to detect change at 80% power and 5% significance, 16 participants would be needed. With respect to glutamate, we found no studies to help determine what should be the expected change in the concentration of glutamate after acute pain. As a second-best option, we used the results of a recent study on salivary glutamate in people with chronic pain (migraine) where this was compared to salivary glutamate in healthy controls (*Nam et al., 2018*). Using data conversion to mean and standard deviation (SD) (*Abbas, Hefnawy & Negida, 2024*) and then applying the paired $t$-test, a sample size of 18 would be sufficient for a paired before-after study with 80% power and 5% significance (*Dhand & Singh, 2025*). On this basis, a sample size of 18 participants was considered sufficient for this investigation.

## Experimental design and protocol
### Procedures

Participants were asked to refrain from eating food and drinking anything except water for at least 3 h before the experiment and to brush their teeth more than one hour before the start of the experiment to remove any food residue from their mouths. There were no dietary restrictions. All experiments took place in the afternoon in a 2-hour window between 14:00 and 17:30 on separate days to ensure the participants had passed the post-lunch peak in their cortisol levels.

### Pain induction

The CPT was used in this study due to its simplicity, controllability, and effectiveness in inducing pain over a short period of time (*Mitchell, MacDonald & Brodie, 2004*; *Lamotte et al., 2021*). Participants submerged their non-dominant forearm and hand into an ice

bath (at 0–5 °C) for as long as they were able to endure or up to a maximum of 5 min. Participants had full control over when to start and end the immersion of their arm in the ice bath. A researcher stayed with the participants during CPT sessions ensuring that the study protocol was followed including the collection of saliva samples. The temperature of the water bath was continuously monitored with a thermometer placed inside it.

### Pain intensity measures

Participants were asked to record the maximum pain intensity they experienced during the CPT on a 10 cm visual analogue scale (VAS) immediately after removing their arm from the ice bath. Participants' heart rate and blood pressure were monitored every 5 min for a full 1 h (but not coinciding with CPT). An experimenter stayed with the participants during the CPT and would ask for verbal confirmation of the participant's well-being before, once during, and after the experimental session.

### Saliva collection and concentration measurements

At the beginning of the session, participants were asked to rest comfortably in a chair for 10 min to adapt to the experimental setting. They then provided two preliminary saliva samples. The initial saliva sample was collected 30 min prior to the start of the CPT. The second saliva sample was collected 15 min after the initial sample. Saliva samples were collected immediately after the termination of CPT ($t = T_{CPT}$) and then at 10, 20, 30, 40, 50 and 60 min. From a prior literature review, we expected the rise and fall of cortisol to be within a 40-minute time frame (*Zarnegar et al., 2024*). On this basis, measurements for cortisol were done at 10, 20, 30, 40 and 60 min. Glutamate was measured on all the samples including the sample taken at 50 min.

At each time point, saliva samples of 0.5–1.0 ml volume were collected in cryovials using saliva collection aids (Salimetrics LLC, State College, PA, USA). To collect whole saliva, we used the passive drool method (*Jasim et al., 2016*). The samples were kept on ice during the experiment to prevent the degradation of peptides. After obtaining all the saliva samples (that is up to a maximum of 1 h after the end of the experiment), the samples were frozen at −20 °C until assayed.

Salivary cortisol and glutamate measurements were done at the same time and on the same samples. The levels of biomolecules in the collected saliva samples were measured by optical techniques using a microplate reader (Infinite M200 PRO; Tecan, Shanghai, China). Glutamate levels were measured using glutamate assay kits (ab83389) by Abcam (Cambridge, UK). Cortisol levels were measured using high-sensitivity immunoassay kits (1-3002-SAL) by Salimetrics (State College, PA, USA). All measurements were done in duplicates as specified by the manufacturers of the kits and the average value of the two was taken and reported as data points.

## Statistical analysis

GraphPad Prism 8.0.1 (GraphPad Software Inc., LaJolla, CA, USA) was used for data analysis. Data were checked for normality of distribution using the D' Agostino and Pearson test and presented as median and interquartile range (IQR). As well as the absolute values of concentration measured for each biomolecule, the percentage change in the

biomolecule levels relative to the baseline are also reported in this paper. The Friedman test was used followed by Dunn's multiple comparisons test to compare the levels of each biomolecule between saliva samples collected at different time points. The Friedman test is a non-parametric statistical test for comparing related values and Dunn's multiple comparisons test was used to allow exact pairwise comparisons. Gender influences on the concentration levels were evaluated using unpaired $t$-test with Welch's correction for normally distributed data and the Mann–Whitney U test for non-normally distributed data. Welch's correction was used to avoid the assumption of equal variance in the compared samples. Spearman correlation coefficients were calculated to assess the relationship between the two biomolecules and the VAS or CPT duration. All tests were two-tailed and a $p$-value of less than 0.05 was used as an indicator of statistical significance.

# RESULTS

## Demographic characteristics

Eighteen healthy adult individuals (nine males and nine females, according to sex assigned at birth) were recruited to the study. Tests of normality showed that the data are not normally distributed. The demographic characteristics of the participants are summarised in Table 1. The median age of all participants was 25 years, with a range of 21–40 years. The CPT duration ($T_{CPT}$) for all the participants was 5 min (the maximum duration of the test) except for one male participant who removed his arm at 1.33 min after the start of the test. The most frequently reported VAS pain intensity score among the participants was 6. The range of pain intensity on the VAS reported by male participants was 3–8 with a median value of 6, while the range of VAS scores reported by female participants was 6–8 with a median value of 6.5. No statistically significant differences with respect to age, CPT duration and VAS were found between male and female participants (two-tailed Mann–Whitney test).

## Changes in biomolecule levels

The measured individual values of salivary glutamate and cortisol concentration are shown in Figs. 1A and 1B. The medians and the IQRs of the biomolecules' concentrations are reported in Table 2. There was no statistically significant difference between the concentration values at the two time points prior to CPT for either cortisol or glutamate. On this basis, the values at −30 and −15 min were averaged and used as the baseline concentration value.

### Glutamate

The baseline glutamate level (median = 4.90, IQR = 4.7 ng/µL) was slightly above the upper limit of the normal range of salivary glutamate reported in the literature (1.47–4.41 ng/µL) (Nam et al., 2018). Immediately after the CPT, the median glutamate levels increased and then dropped below the baseline at $t = +10$ min. The glutamate levels fluctuated thereafter, rising at $t = +20$ min compared to baseline, and then dropping again at $t = +50$ min to significantly lower than the baseline levels (Friedman's test with Dunn's adjusted *post hoc* test: $p = 0.014$). The $t = +50$ min time point was the only time when there was a statistically significant difference with the baseline level.

**Table 1  Medians and ranges of participants' age, CPT duration, and VAS pain scores for both sexes.**

| Characteristic | Male (n = 9) | Female (n = 9) | p-value |
|---|---|---|---|
| Age (years) | 23 (21–40) | 24 (21–40) | 0.85 |
| $T_{CPT}$ (min) | 5 (1.33–5) | 5 | 0.47 |
| VAS | 6 (3–8) | 6.5 (6–8) | 0.49 |

**Table 2  Measured levels (medians and interquartile ranges (IQR)) of glutamate and cortisol across the different time points during the experimental session.** Differences that were statistically significant between all the time points for each biomolecule are indicated with the same superscript.

| Time points (min) | Glutamate | | | Cortisol | | |
|---|---|---|---|---|---|---|
| | Medians (IQR) | | | Medians (IQR) | | |
| | (ng/µL) | | | (µg/dL) | | |
| | Female | Male | All | Female | Male | All |
| Bas | 4.86 (4.7) | 4.50 (4.8) | 4.90 (4.7)[*] | 0.10 (0.09) | 0.16 (0.11) | 0.14 (0.1)[**] |
| $T_{CPT}$ | 5.85 (6.0) | 5.37 (4.9) | 5.66 (4.6) | 0.13 (0.1) | 0.18 (0.2) | 0.16 (0.2) |
| +10 | 1.82 (2.1) | 2.59 (2.7) | 2.55 (1.7) | 0.23 (0.2)[i] | 0.56 (0.7)[i] | 0.34 (0.4)[**] |
| +20 | 7.28 (5.5) | 6.40 (6.8) | 6.84 (6.7) | 0.17 (0,2)[ii] | 0.55 (1,0)[ii] | 0.29 (0.7) |
| +30 | 1.54 (3.9) | 2.60 (2.6) | 2.31 (3.1) | 0.18 (0.2)[iii] | 0.44 (0.7)[iii] | 0.28 (0.6) |
| +50 | 1.43 (4.3) | 2.61 (2.9) | 2.08 (3.3)[*] | – | – | – |
| +60 | 7.98 (8.1) | 7.45 (7.3) | 7.55 (7.5) | 0.10 (0.3) | 0.23 (0.1) | 0.20 (0.2) |

Notes.

[*]$p < 0.05$.

[**]$p < 0.01$

Differences that were statistically significant between the sexes for each time point are indicated with the same superscript.

[i, ii, iii]$p < 0.05$.

The differences between the values that are not accompanied by a superscript were not statistically significant.

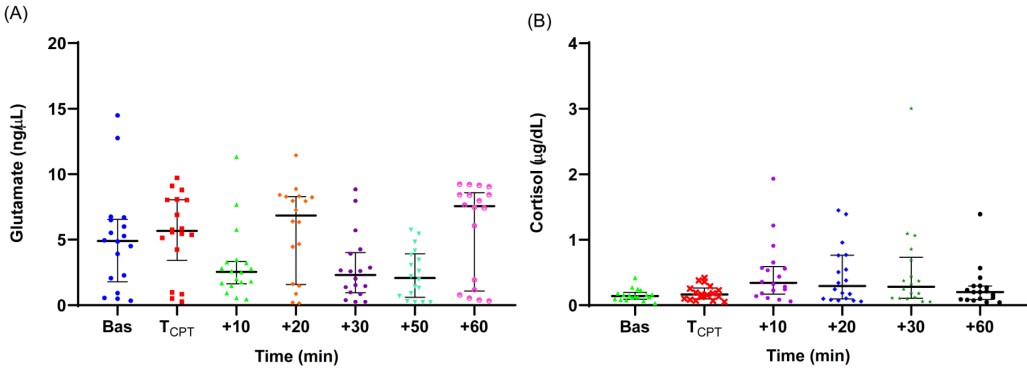

**Figure 1  Individual salivary glutamate (A) and cortisol (B) concentration values during the experiment.** Medians are shown with horizontal lines. The interquartile ranges (IQR) are presented with vertical lines.

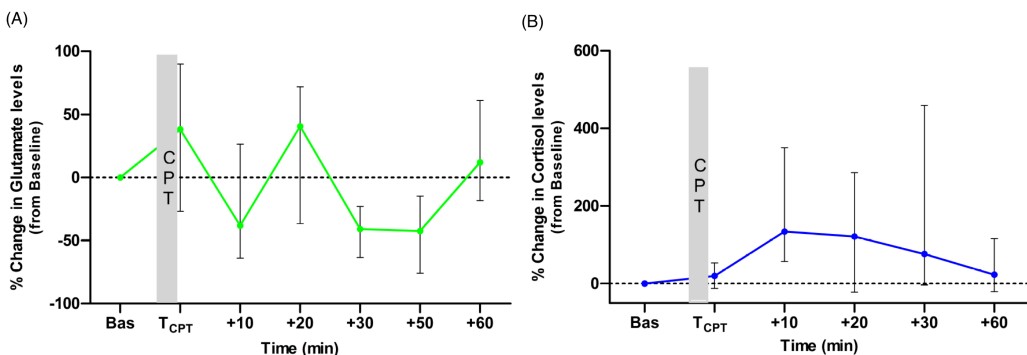

**Figure 2** Medians of the normalised (to the baseline levels) percentile changes of glutamate (A) and cortisol (B) levels. The interquartile ranges (IQR) are presented with vertical lines.

### Cortisol

A significant increase from the baseline (median = 0.14, IQR = 0.1 µg/dL) was observed immediately after the CPT. The median salivary cortisol concentration at $t = +10$ min was significantly greater than the baseline level (Friedman's test with Dunn's adjusted *post hoc* test: $p = 0.007$). Cortisol levels at $t = +10$ min were also significantly higher than the levels at t $= T_{CPT}$ and $t = +60$ min ($p = 0.03$ and $p = 0.02$, respectively), but not significantly higher compared with $t = +20$ and $t = +30$ min, indicating that cortisol levels peaked 10–20 min after the CPT.

The medians of percentage change from baseline for the two biomolecules at each time point are presented in Figs. 2A and 2B. Glutamate levels increased from the baseline by 38% (median) immediately after the CPT. The median percentage change values were higher than the baseline levels also at $t = +20$ and $t = +60$ min but not at $t = +10$, $t = +30$ and $t = +50$ min, (Fig. 2A). For cortisol, the median percentage change was higher than the baseline at all time points. Cortisol levels in 15 out of 18 participants increased from the baseline at $t = +10$ min. The normalised levels gradually reduced to within 23% of their baseline by the end of the experiment (see Fig. 2B).

### Correlation analysis with pain-VAS scores and CPT duration

We found a moderate positive correlation between the VAS scores during CPT and the percentage change in glutamate levels at $t = T_{CPT}$ (Spearman correlation coefficient $r = 0.51$, $p = 0.02$). For cortisol, a moderate positive correlation between the VAS scores and the percentage change in cortisol was found at $t = +20$ and $t = +30$ min (Spearman correlation coefficient ($r = 0.52$, $p = 0.02$ and $r = 0.54$, $p = 0.01$, respectively). There was no correlation between the duration of the CPT and the normalised levels of glutamate or cortisol at any time point (Spearman correlation coefficient, $p > 0.05$).

### Sex differences

The changes in the levels of biomolecules in male and female participants are shown in Figs. 3A and 3B. The pattern of change in salivary glutamate was similar in male and female subjects with no statistically significant difference between the sexes (*t*-test with

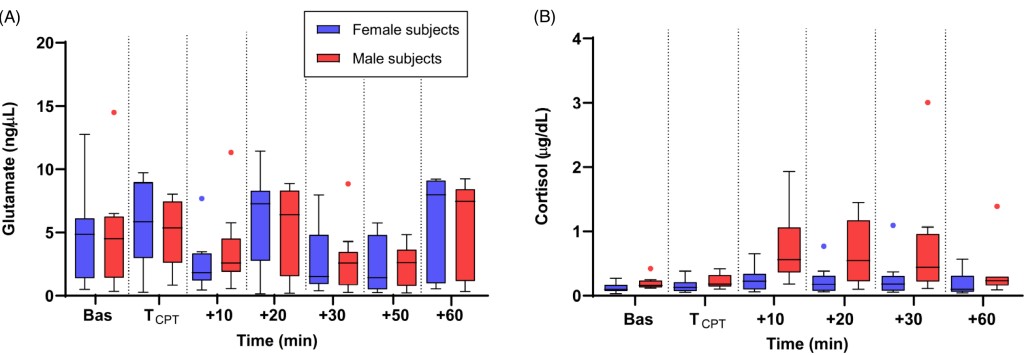

**Figure 3** Medians of salivary glutamate (A) and cortisol (B) concentrations in the male (red colour line) and female (blue colour line) groups. The interquartile ranges (IQR) are presented with vertical lines.

Welch's correction and Mann–Whitney U test, $p > 0.05$) at any of the time points. For cortisol, male participants had a greater increase in cortisol concentration at $t = +10$, $+20$ and $+30$ min compared to females and these differences were found to be statistically significant (Mann–Whitney U test, $p = 0.01$, $p = 0.02$, and $p = 0.04$, respectively).

## DISCUSSION

We have investigated the salivary concentrations of two nociception-related biomolecules, glutamate and cortisol, after experimentally induced acute cold pain in healthy volunteers. Salivary cortisol increased after CPT, reaching its peak 10–20 min after the completion of the task and returning to near baseline after CPT termination. Salivary glutamate levels fluctuated after CPT but none of the changes were statistically significant except at $t = +50$ min when there was a drop below baseline. To our knowledge, this is the first report of an investigation on change in salivary glutamate concentration after an episode of acute pain.

To date, cortisol is the most studied salivary biomolecule after an episode of pain. In this study, there was a rise in salivary cortisol after acute pain with a pattern that is consistent with the results of previous studies on this topic (*Zarnegar et al., 2024*; *Bachmann et al., 2018*; *Goodin et al., 2012*; *Nakajima & al'Absi, 2011*; *Finke et al., 2021*). As with most studies in this field, salivary changes have been studied in highly controlled conditions where healthy young adult participants with no pre-existing painful problem were exposed to a specific noxious stimulus. This limitation was self-imposed because there is no information on change in salivary glutamate after acute pain in the literature. We therefore found it important to understand the profile of change in healthy people in specified conditions in the first instance before studying it in the general population. On the other hand these limitations self-evidently impact the generalisability of the findings and their relevance to clinical practice. If salivary biomolecules are to be used as useful biomarkers, a nuanced understanding of the factors that could be involved in their variance is needed. We therefore added analyses to compare the pattern of biomolecule change in
males and females. A similar pattern of change was observed in both sexes, for both cortisol and glutamate. For cortisol, however, male participants had higher responses than females and the differences between them were statistically significant at multiple time points. A possible explanation for this may be differences in HPA response patterns to stress between the sexes (*Hellhammer, Wüst & Kudielka, 2009*). However, it is important to acknowledge that this is not a consistent finding in the current literature on cortisol and cold pain (*Zarnegar et al., 2024*) and a suitably powered study is needed to determine whether there are true differences. Noxious stimuli are powerful activators of the stress response. The observations of this study and those similar to it may therefore be due to a surge in autonomic endocrine processes rather than the activation of nociceptive pathways per se. As with most studies in this field, salivary changes have been studied in a group of healthy young adults with no pre-existing painful conditions who were exposed to a specific controlled noxious stimulus. This limitation was self-imposed because there is no information on the change in salivary glutamate levels after the induction of acute pain in the literature. With respect to cortisol release, this study was observational and not designed to show causal relationships. Noxious stimuli are powerful activators of the stress response. The observations of this study and those similar to it could therefore be due to a surge in autonomic endocrine processes rather than due to the activation of nociceptive pathways per se.

Most research on salivary biomolecules has been conducted on healthy participants (*Zarnegar et al., 2024*), where researchers have stated that volunteering participants were healthy at the time of recruitment, giving no evidence of further assurance sought by direct questioning. We asked participants if they felt well directly, and, asked an additional question with respect to whether they required regular medication as a confirmatory health question. These questions rely on self-reporting. Current evidence suggests that although inevitably subjective, self-reported health does correlate with objective health measures at least in Western societies (*Lorem et al., 2020*). However, it is important to acknowledge that the use of a validated health assessment questionnaire should be considered in future studies to avoid bias related to confounding variables.

In the last 20 years, technologies in the detection of salivary components have progressed rapidly, allowing the measurement of a wide range of biomolecules (*Song et al., 2023*). However, finding biomolecules with a consistent and reliable relationship to acute or chronic pain so that they can be considered clinically useful biomarkers has proven elusive. While the consistency in the pattern and timing of change in cortisol levels after CPT reported by various investigators is encouraging, there are other complexities in the use of cortisol in particular its circadian rhythm of production in the body (*Kirschbaum et al., 1999*), the menstrual cycle in females (*Womack, MacDermott & Jessell, 1988*), and the impact of chronic stress or chronic illness (*Goodin et al., 2012*; *Uhart et al., 2006*) on the HPA axis. These are significant limitations to using cortisol alone as a marker for acute nociception not only because greater reliability is needed for titration of analgesic medication, but also because the time scale of change, typically reaching a peak 10 to 20 min after an episode of acute pain, is not rapid enough for useful analgesia titration. In this study, we also chose to look at salivary glutamate and found that the levels fluctuated after

CPT with no significant change from baseline, and therefore, no trend could be observed either. We have found no other studies on glutamate levels after experimental induction of acute pain, however an important limitation of this work is that only one pain modality (cold pain) has been considered.

It is possible that we missed a peak in glutamate concentration in the first 10 min after the CPT due to the saliva sampling time points in our experimental design. The question is whether salivary glutamate deserves further attention with measurements that are taken at time points closer to the time of pain induction or in experiments that induce other modalities of acute pain. Since glutamate is released both centrally and in the peripheral tissues after noxious stimuli, a hypothesis that there would be a salivary rise after inducing acute pain is reasonable. However, in this context, the mechanism through which salivary glutamate would rise in the saliva deserves some consideration. This may be secondary to a rise in blood concentration after a period of intense glutamate release from nerve endings, which then diffuses into the saliva. Arguing against this, in the literature on chronic pain conditions, the relationship between high salivary glutamate and pain intensity has been inconsistent and there is little correlation between salivary and plasma glutamate levels (*Jasim et al., 2020*).

## CONCLUSION

On balance, the findings in this study taken in combination with existing studies in chronic pain, would suggest that glutamate may not be a good candidate as a salivary biomarker in acute pain. Nonetheless, considering how valuable salivary detection of nociceptive activity could be in clinical practice, it would be useful to design similar experiments with additional pain modalities and with measurement of glutamate at time points closer than 10 min after pain induction.

### Funding

This work was funded by the Wellcome Trust (Grant number: 204841/Z/16/Z). The funders had no role in study design, data collection and analysis, decision to publish, or preparation of the manuscript.

### Grant Disclosures

The following grant information was disclosed by the authors:
The Wellcome Trust: 204841/Z/16/Z.

### Competing Interests

The authors declare there are no competing interests.

### Author Contributions

- Roxaneh Zarnegar conceived and designed the experiments, prepared figures and/or tables, authored or reviewed drafts of the article, and approved the final draft.

- Angeliki Vounta analyzed the data, prepared figures and/or tables, authored or reviewed drafts of the article, and approved the final draft.
- Arisara Amrapala performed the experiments, analyzed the data, prepared figures and/or tables, authored or reviewed drafts of the article, and approved the final draft.
- Sara S. Ghoreishizadeh conceived and designed the experiments, authored or reviewed drafts of the article, and approved the final draft.

## Human Ethics

The following information was supplied relating to ethical approvals (i.e., approving body and any reference numbers):

The University College London Ethics Committee granted Ethical approval to carry out the study within its facilities (Ethical Application Ref: 15021/001).

## Data Availability

Raw data is available as a Supplemental File.

## Supplemental Information

Supplemental information for this article can be found online at http://dx.doi.org/10.7717/peerj.19625#supplemental-information.

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
