# Peer review of "A study of salivary cortisol and glutamate after the cold pressor task in healthy adults"

_PeerJ, doi:10.7717/peerj.19625_

## Round 0.1 · original submission · Major Revisions

Dear authors,

Thank you for your submission. Please, refer to the reviewers' comments and answer point-by-point to the reviewers' comments / questions.

·

Basic reporting

The manuscript generally uses clear and professional English, but there are a few areas that could be improved:
In the abstract, "CPT (median 0.34 ng/¿L, IQR=0.4, p=0.007)" should be "CPT (median 0.34 ng/µL, IQR=0.4, p=0.007)" to correct the unit symbol.
The use of apostrophes for time points (e.g., "t= +50'") is inconsistent and should be standardized throughout.
The literature references and background context are sufficient, providing a good foundation for the study. The article structure is professional, with clear sections for introduction, methods, and results. However, the results section is missing from the provided manuscript, which is a significant omission.
Figures are not included in the provided text, making it difficult to assess their quality and relevance. The authors should ensure all figures are properly labeled, described, and referenced in the text.
There is no mention of raw data sharing in the manuscript. The authors should explicitly state whether and how the raw data will be made available, in line with PeerJ's data sharing policy.

Experimental design

The study appears to be original primary research within the scope of PeerJ. The research question is well-defined and relevant, investigating changes in salivary cortisol and glutamate levels following acute cold pain induction.
The methods are described in sufficient detail to allow replication. However, some improvements could be made:
The specific model of equipment used for measuring cortisol and glutamate concentrations should be mentioned.
More details on the statistical analysis methods would be beneficial.
The ethical standards appear to be high, with appropriate approval and consent procedures described.
Validity of the Findings
As the results section is missing from the provided manuscript, it is impossible to fully assess the validity of the findings. This is a critical issue that needs to be addressed before publication.
Based on the information provided in the abstract:
The statistical analysis appears sound, with appropriate use of median and IQR for non-normally distributed data.
The conclusions seem to be well-stated and linked to the original research question. However, without the full results section, it's impossible to verify if they are limited to supporting results.
The authors should ensure that all underlying data are provided and that the statistical analyses are robust and controlled.

Validity of the findings

The manuscript fails to meet some of standards in the following areas:
Data availability: There is no explicit statement about the availability of underlying data. The authors should clearly state how and where the raw data will be made available, in line with PeerJ's data sharing policy
.
Results and conclusions: The results section is missing from the provided manuscript, making it impossible to assess if the data are robust, statistically sound, and controlled. Similarly, without the results and discussion sections, it's not possible to evaluate if the conclusions are well-stated, linked to the original research question, and limited to supporting results
.
Statistical analysis: While the abstract mentions some statistical results, more details on the statistical methods used would be beneficial. This information should be included in the methods section
.
Replication rationale: The manuscript does not explicitly state the rationale and benefit to literature for this study. While the introduction provides some context, a clearer statement of how this research fills an identified knowledge gap would strengthen the paper

Additional comments

General comments:
The results section needs to be included in the manuscript.
Figures should be provided and properly referenced in the text.
A statement on data availability should be added.
The discussion section (if present) should critically evaluate the findings in the context of existing literature.

Reviewer 2 ·

Basic reporting

1.The introduction lacks a clear explanation of the relationship between cortisol and glutamate. Please elaborate on how these two are interconnected to strengthen the context and provide a more comprehensive background for the study.
2. The manuscript does not address the importance of investigating gender differences. Please provide a rationale for why examining these differences is relevant and significant to the study.
3.Salivary cortisol measurements were done at the same time and on the same samples. lines 59-60. This sentence should be included in the methodology section.
4.To our knowledge, this is the first report of an investigation on change in salivary glutamate concentration after an episode of acute pain. lines 60-62. This sentence should be included in the discussion section.

Experimental design

1. The current sample size (9:9) is very small, which greatly limits the statistical power to detect differences. Consider increasing the number of participants to enhance the reliability and robustness of the results.
2. This study examines 2 biomolecules, but in the sentence on line 139, it states 3 biomolecules.
3. In the methodology section, it is recommended to specify a minimum time criterion for participants who are unable to immerse their hand and arm in cold water as required. Participants who do not meet this criterion should be excluded from the study to ensure consistency and reliability (lines 149–150).
4. Please clarify the criteria used to define healthy participants. Is a medical examination conducted, or is a health assessment questionnaire employed for this purpose? This approach raises concerns about the validity of the baseline health assessments, potentially introducing bias and confounding variables.
5.How the temperature was maintained, and how compliance with the protocol was ensured.

Validity of the findings

Result
1.Please verify the unit of the biomolecule in the text and in the results.
2. Please clarify why the cortisol levels are not observed during the 50-minute period following the CPT. It would be helpful to provide an explanation for this absence of effect.
Conclusion
This study lack of conclusion. The study would benefit from a conclusion that directly addresses the research objectives, as this would help readers better understand the key findings and their implications.

Additional comments

The study presents several limitations that have not been addressed by researchers. The following additional limitations should also be considered.

Reviewer 3 ·

Basic reporting

Dear Authors,

Thank you for this interesting research. It is well-structured and professionally written. References are up-to-date and relevant to the field of study.

Experimental design

This is an observational study with a clear aim and hypothesis. The authors have obtained ethical approval. The methods are sufficiently described, and the statistical analysis is valid.

Validity of the findings

However, I believe the sample size of 18 participants is too small, which reduces the generalizability of the results. Additionally, including only healthy individuals eliminates the possibility of intergroup comparisons. Furthermore, only one quantitative sensory test (QST) was performed—the cold pressor test (CPT).

Taking this into account, I feel this research lacks sufficient scientific relevance. Increasing the number of participants and incorporating additional QSTs would significantly improve the validity of these findings.

Although the article is well written in its current form, I believe it does not fully align with the scope of this journal. Expanding the study sample to include more subgroups and additional tests would greatly enhance the article, making it a highly valuable contribution to the literature.

---

## Round 0.2 · accepted · Accept

Dear authors,

The revisions have been satisfactory. I am now accepting your manuscript for publication. Congratulations!

Reviewer 2 ·

Basic reporting

The manuscript is generally well-written, demonstrating clear and professional use of English. The rationale behind the study is more coherently presented, with improved articulation of the sources and their significance. The underlying assumptions are now more clearly stated and easier to follow. Additionally, The manuscript now includes a concise summary that effectively reflects the study's objectives, enhancing the overall clarity and focus of the introduction.

Experimental design

The sample size (9:9) is very small. However, the author has shown how to calculate the sample, which makes the research process more concise.

Validity of the findings

No comment